# PingPong: a Benchmark for Role-Playing Language Models with User Emulation and Multi-Model Evaluation

## Abstract

We introduce a benchmark for evaluating the role-playing capabilities of language models. Our approach leverages language models themselves to emulate users in dynamic, multi-turn conversations and to assess the resulting dialogues. The framework consists of three main components: a player model assuming a specific character role, an interrogator model simulating user behavior, and several judge models evaluating conversation quality. We conducted experiments comparing automated evaluations with human annotations to validate our approach, demonstrating strong correlations across multiple criteria. This work provides a foundation for a robust and dynamic evaluation of model capabilities in interactive scenarios.

## 1 Introduction

Language models, which predict plausible language, have dominated natural language processing since BERT (Devlin et al., 2019), with models like ChatGPT (Ouyang et al., 2022) showcasing advanced conversational capabilities.

In this paper, we focus on role-playing language models for entertainment purposes. These models are assigned specific characters or personas and are tasked with maintaining these roles while engaging and entertaining users. While there are other important applications of role-playing language models, such as training mental health specialists (Louie et al., 2024) or simulating human opinion dynamics (Chuang et al., 2024), they are beyond the scope of this paper.

We introduce a novel benchmark for evaluating role-playing language models. From our experience with language models, we believe direct interaction is the most effective way to assess a language model's conversational abilities. However, humans often lack time to test new models manually, and many popular benchmarks are limited to single-turn interactions (Dubois et al., 2024a; Hendrycks et al., 2021). These benchmarks are also becoming less reliable due to test data contamination (Deng et al., 2024). To address this, we propose using language models to emulate users in role-playing conversations and automatically evaluate the resulting dialogues.

Our methodology, illustrated in Figure 1, involves three key components: a player model assuming a character role, an interrogator model simulating user behavior, and a judge model evaluating conversation quality. Our work builds upon existing benchmarks, such as EQ-bench (Paech, 2023), introducing an approach to role-playing evaluation.

Our contributions:

- We propose a benchmark for assessing the role-playing abilities of language models. The combination of the following traits makes it novel:
  - **Multi-turn**: All conversations have multiple turns to be closer to the real usage of role-play models.
  - **Dynamic**: Interrogator questions are generated by language models with sampling and are not pre-defined. Each evaluation run produces different questions, making it harder for models to memorize responses to make test data contamination harder.

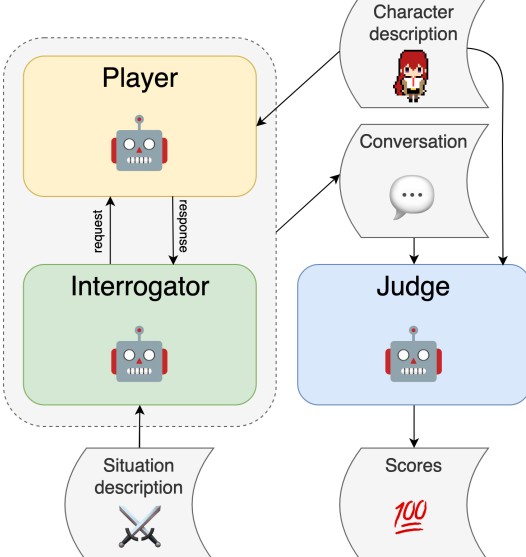

Figure 1: This diagram illustrates the flow of interactions in the proposed benchmark. There are three main components with different language models: a player, an interrogator, and a judge. The player assumes some character role, the interrogator acts as a user in a specific situation, and the judge evaluates final conversations.

  – **Multi-model**: There are several judges to mitigate individual model biases and have a better correlation with humans. It also allows evaluation of the models that are used as judges.
- We validate our benchmark through correlation with manual annotations and comparison with other benchmarks.
- We show that the multi-model setup correlates better with humans than a single model.
- We discover that fine-tuning models for creative writing improves their role-playing abilities.

All the results, prompts, and scripts are available online[1]. The benchmark website has the final up-to-date leaderboards [2] and all the conversations with example-wise scores. It is available for English and Russian languages.

## 2 RELATED WORK

**Role-play capabilities and evaluation.** Various commercial services exploit role-play abilities of language models, including Character.ai[3] and Chai (Irvine et al., 2023). There are academic and community attempts to create similar systems with open datasets, code, and models, such as PIPPA, ChatHaruhi, Character-LLM (Gosling et al., 2023; Li et al., 2023; Shao et al., 2023), MythoMax[4], or Magnum[5]. Several static benchmarks for role-playing exist, including ECHO, InCharacter, and CharacterEval (Ng et al., 2024; Wang et al., 2024; Tu et al., 2024).

PersonaGym (Samuel et al., 2024) is close to our work, featuring dynamic question generation based on the environment ("situation" in our terminology) and the currently selected persona. There is also a very similar dynamic benchmark, RPBench-Auto[6]. It is based on the same assumptions and features and has a structure similar to one of the versions of our benchmark, which is surprising since

---

[1]https://github.com/AnonResearch01/ping_pong_bench
[2]https://anonresearch01.github.io/ping_pong_bench/
[3]https://character.ai/
[4]https://huggingface.co/Gryphe/MythoMax-L2-13b
[5]https://huggingface.co/anthracite-org/magnum-v2-123b
[6]https://boson.ai/rpbench-blog/

this benchmark was developed completely independently of our work. The major difference from our work is that evaluation is based on side-by-side comparisons with the baseline model, while we produce single-point evaluations.

A different approach to evaluation would be using online metrics, such as retention rates or user ratings (Irvine et al., 2023). However, this approach is only viable if you already have a product with a substantial user base.

**Automatic and multi-model evaluation.** LLM-as-a-Judge (Zheng et al., 2023) is an evaluation method that relies on language models, such as GPT-4, instead of humans. Popular benchmarks using this method include AlpacaEval, EQ-bench, Creative Writing, and BiGGen Bench (Dubois et al., 2024a; Paech, 2023; Kim et al., 2024). The validity of these benchmarks relies on their high correlation with human annotations, specifically with Chatbot Arena (Chiang et al., 2024).

However, all these benchmarks rely on a single model as a judge, which may introduce various biases, including the self-evaluation bias (Panickssery et al., 2024; Xu et al., 2024). PoLL (Verga et al., 2024) authors aggregate evaluations from different language models in a similar way we do, with average pooling. They show that ensembling different models for evaluation increases correlation with human annotations. There is also another more agentic approach (Chan et al., 2023) with a referee team.

**Multi-turn evaluation and data contamination.** Most benchmarks are single-turn, which contrasts with the real-world usage of language models. There are multi-turn benchmarks, such as MT-Bench-101 (Bai et al., 2024) and MT-Eval (Kwan et al., 2024), though they focus on specific capabilities, and their evaluation procedures still differ from how humans implicitly rate language models.

Another major problem for the static public benchmarks is data leakage into the pre-training datasets of language models (Deng et al., 2024). It's challenging to avoid contamination since such tests are usually stored online and considered "code" during pre-training. This can occur even without malicious intent from model creators. The most obvious solution is to close benchmarks completely, which requires trusting benchmark organizers, which is difficult in a highly competitive environment. Alternative solutions include regularly updating benchmarks with new test data (White et al., 2024) or dynamically generating test data using existing language models.

# 3 METHODOLOGY

## 3.1 ROLE DEFINITIONS

Our framework comprises three principal roles: player, interrogator, and judge, inspired by the Turing test (Turing, 1950). However, our approach differs in the number of agents, the player's objective, and the use of machine-based interrogators and judges.

Language models can take three possible roles.

- **Player** assumes the role of a specific character based on a provided character card.
- **Interrogator** engages with the player within a given situation or towards a specific goal, simulating user behavior.
- **Judge** evaluates the player's responses against predetermined criteria.

Role assignments are implemented through a combination of system and user prompts. We use only models that support chat templates. For models lacking dedicated system prompts, such as Gemma 2 (Gemma, 2024), all instructions are incorporated into the user prompt.

This setup is **asymmetrical** since the player only gets the character description while the interrogator only gets the situation information. This is intentional, as typical use cases of role-playing models are asymmetrical. However, it is possible to modify it to make it symmetrical by providing character descriptions and situations both to the player and the interrogator. Symmetrical setups might be useful in other domains.

## 3.2 JUDGE

The scoring is single-point, with no reference examples or pairs. The judge used three main evaluation criteria:

- **Character consistency**: The player's answers align perfectly with the assigned character; they correspond to the character's description.
- **Entertainment value**: The player's responses are engaging and entertaining.
- **Language fluency**: The player's language use is of the highest quality, without any mistakes or errors. The player is perfectly fluent.

These criteria reflect the main things we expect from the model during role-playing. In addition to them, we ask whether the player refused to answer.

We prompt a model to explain itself before giving a score, using quotes from the conversation. It must also return a set of scores for every turn of the conversation.

## 3.3 VERSION 1: COMBINED INTERROGATOR AND JUDGE

In the initial version, the roles of interrogator and judge were merged. This combined entity receives the player's character card, a situational context, and a list of evaluation criteria. It evaluates the player's most recent response and generates the subsequent user utterance.

We selected Claude 3.5 Sonnet as the interrogator/judge model based on the Judgemark[7] results, hypothesizing a correlation between creative writing and role-play capabilities. The evaluation uses a 10-point scale for every criterion.

The key issues of this approach are:

- **Unrealistic user emulation:** In many real-world use cases, users lack complete information about character profiles, and to correctly emulate it, we should not provide complete character information to the interrogator.
- **High costs:** The task of the interrogator is much easier than the task of the judge, so it doesn't make sense to use the same expensive model for both of them.
- **Non-optimal decoding strategies:** Some decoding strategies are good for judgment but not for interrogation. For instance, a higher temperature benefits the interrogator but not the judge.

## 3.4 VERSION 2: SEPARATED ROLES AND MULTI-MODEL EVALUATION

Recognizing the limitations of the combined approach, we developed a second version with distinct interrogator and judge roles. It allows flexible control of costs and information flow.

Furthermore, we identified the inadequacy of single-model evaluation. To address this, we implement a multi-model evaluation system. This approach involves averaging scores from different judge models. In this particular setup, we used Claude 3.5 Sonnet and GPT-4o, the top two models, by correlation with manual annotations. We tried several more sophisticated approaches, but the average worked best.

As an interrogator, we take GPT-4o Mini. According to the version 1 leaderboard (still available online), it has the same generation quality as GPT-4o but is cheaper. This version uses a 5-point Likert scale to match human annotations instead of a 10-point scale.

## 4 EXPERIMENTS

### 4.1 CORRELATION WITH HUMAN ANNOTATIONS

First, we verified that the proposed judges correlate well with human evaluations. Using the version 1 setup, we created 64 conversations for each of more than 13 language models. Then, we sampled

---

[7]https://eqbench.com/judgemark.html

250 and 265 samples for English and Russian, respectively, and manually annotated them using a 5-score Likert scale.

The annotation was performed by five native Russian speakers with diverse academic and professional backgrounds who were proficient in English. After reading each sample, annotators answered three questions corresponding to three metrics. We averaged scores between annotators for every sample and every metric. The details of the annotations process can be found in Appendix B

Then, we computed the Spearman correlation (Spearman, 1904) between aggregated manual scores and automatic annotations from different setups. We chose rank correlation because scales were different in versions 1 and 2, and we wanted to compare them.

Calculating metrics for version 1 and models different from Claude 3.5 Sonnet is impossible since version 1 uses a combined interrogator and judge, so we can't get new scores for existing conversations.

## 4.2 LEADERBOARDS

We calculated automatic metrics across language model families, both proprietary and open-source. For each model, we report the mean scores per metric, proportion of conversations with refusals, overall metric average, and confidence intervals (via bootstrapping) for final metrics.

We evaluate each model using 64 conversations across 8 characters and 8 situations, with varying conversation lengths. The evaluation process is computationally efficient, costing less than $3 per model. Since the judge gives annotations for every turn, the overall number of annotations is not 64 but 288. We do not want to make this sample bigger since it will increase the runtime and costs, and we have budget constraints.

We covered diverse sources in selecting characters and situations, including computer games, TV shows, movies, books, and anime. Situations fall into two categories: common user patterns and attempts to break model behavior. In Appendix C, we estimate the fraction of real user situations covered by our set of situations.

### 4.2.1 LENGTH PENALTY

Both language models and humans exhibit verbosity bias (Dubois et al., 2024b). The longer the output, the higher the chance of being positively evaluated. We use a length penalty similar to the Creative Writing[8] benchmark to account for this. We calculate length-normalized scores for all models, penalizing models with a median length of player messages higher than a global median length.

### 4.2.2 TECHNICAL DETAILS

We utilize OpenAI-like API for all models. Some models are used directly from their providers, some are taken from OpenRouter[9], and some are hosted in different modes with RunPod[10].

We use the same sampling parameters for most players: temperature=0.6, top_p=0.9 (Holtzman et al., 2020). Some models, such as Gemma 2, frequently repeated phrases. We addressed this by increasing the temperature and applying an additional frequency penalty. For the interrogator, we use temperature=0.8 and top_p=0.95; for the judge, we use temperature=0.1 and top_p=0.95.

We try to cover different popular families of models, namely OpenAI GPT (OpenAI et al., 2024), Anthropic Claude, Meta Llama (Dubey et al., 2024), Gemini (Gemini et al., 2024), Gemma (Gemma, 2024), and Qwen (Yang et al., 2024). We also evaluate popular role-play and creative writing models featured in OpenRouter and in the Creative Writing benchmark. We do not use base models, only their chat versions.

---

[8]https://eqbench.com/creative_writing.html
[9]https://openrouter.ai/
[10]https://www.runpod.io/

Table 1: Spearman correlations of different models and setups with human expert annotations for English based on 250 samples. P-values are less than 0.0001, except those marked with an asterisk.

| Model | In-character | | Entertaining | | Fluency | | Final | |
|---|---|---|---|---|---|---|---|---|
| | v1 | v2 | v1 | v2 | v1 | v2 | v1 | v2 |
| Claude 3.5 Sonnet | 0.433 | 0.448 | 0.582 | 0.616 | 0.182* | 0.115* | 0.499 | 0.554 |
| Llama 3.1 70B | – | 0.403 | – | 0.573 | – | 0.116* | – | 0.546 |
| GPT-4o | – | 0.396 | – | 0.541 | – | **0.283** | – | 0.517 |
| GPT-4o Mini | – | 0.348 | – | 0.514 | – | 0.019* | – | 0.467 |
| Claude 3 Haiku | – | 0.251 | – | 0.406 | – | -0.069* | – | 0.349 |
| Avg(Sonnet, 4o) | – | **0.460** | – | **0.646** | – | 0.250 | – | **0.604** |

Table 2: Spearman correlations of different models and setups with human expert annotations for Russian based on 265 samples. P-values are less than 0.0001, except those marked with an asterisk.

| Model | In-character | | Entertaining | | Fluency | | Final | |
|---|---|---|---|---|---|---|---|---|
| | v1 | v2 | v1 | v2 | v1 | v2 | v1 | v2 |
| Claude 3.5 Sonnet | 0.291 | 0.374 | 0.497 | 0.553 | 0.210* | **0.548** | 0.379 | 0.547 |
| GPT-4o | – | 0.424 | – | 0.553 | – | 0.413 | – | 0.550 |
| GPT-4o Mini | – | 0.166* | – | 0.393 | – | 0.225* | – | 0.344 |
| Claude 3 Haiku | – | 0.141* | – | 0.265 | – | 0.021* | – | 0.157 |
| Llama 3.1 70B | – | 0.319 | – | 0.367 | – | 0.031* | – | 0.253 |
| Avg(Sonnet, 4o) | – | **0.435** | – | **0.617** | – | 0.529 | – | **0.612** |

### 4.3 COMPARING TO OTHER BENCHMARKS

We hypothesize a correlation between creative writing and role-play capabilities of language models because both creative writing and role-playing require similar capabilities: maintaining consistent character voices/personas, generating engaging and entertaining content, producing fluent and coherent language, and understanding and working within given constraints.

If our hypothesis is true, there should be a correlation between our benchmark and the Creative Writing benchmark. Since we have scores from both benchmarks for each model, we can directly calculate the Spearman correlation between the rankings.

Another benchmark we compare with is RPBenchAuto[11]. Its scene-based setting is the closest to our work. The major difference is that it utilizes side-by-side comparisons with a baseline model instead of single-point evaluations.

## 5 RESULTS

**Automatic judges correlate with humans**. Spearman correlation of different versions of automatic judges can be found in Table 1 and Table 2. For Russian, the only models that stand out are Claude 3.5 Sonnet and GPT-4o, which produce scores with Spearman correlation higher than 0.5. For English, there is also Llama 70B, which has the same level of correlation for the final score.

Correlations are higher than 0.3 for almost all attributes in the case of multi-model evaluation, which is the last row. The only exception is language fluency in English. There are several reasons for this exception. First, annotators were not native English speakers, so it was hard to catch subtle nuances in fluency. Second, most of the tested methods were already excellent in this aspect. In contrast, most models still struggle with Russian, so there is a moderate correlation there.

**Multi-model setup has a higher correlation with humans**. After averaging the final scores from the two models, the correlation between them is higher than 0.6 for both languages and higher than

---

[11]https://boson.ai/rpbench-blog/

Table 3: Leaderboard for Russian, v2, top-10 models by length-normalized (LN) aggregated score. We provide 95% CI widths only for the final score to make the table more readable. Confidence intervals were calculated with bootstrapping.

| Model name | LN score | Agg. | Ref. ratio | Char. | Fluency | Ent. | Length |
|---|---|---|---|---|---|---|---|
| Claude 3.5 Sonnet | $4.62_{\pm0.07}$ | 4.68 | 0.30 | 4.80 | 4.80 | 4.44 | 388 |
| Gemini Pro 1.5 002 | $4.51_{\pm0.09}$ | 4.52 | 0.00 | 4.70 | 4.79 | 4.06 | 223 |
| Gemini Pro 1.5 | $4.49_{\pm0.08}$ | 4.49 | 0.02 | 4.60 | 4.75 | 4.13 | 213 |
| GPT-4o Mini | $4.48_{\pm0.06}$ | 4.49 | 0.00 | 4.62 | 4.82 | 4.04 | 329 |
| GPT-4o | $4.47_{\pm0.08}$ | 4.47 | 0.02 | 4.61 | 4.82 | 3.99 | 301 |
| Qwen 2.5 72B | $4.45_{\pm0.07}$ | 4.46 | 0.02 | 4.55 | 4.80 | 4.02 | 326 |
| Gemma 2 Ataraxy 9B | $4.45_{\pm0.07}$ | 4.45 | 0.00 | 4.61 | 4.52 | 4.21 | 302 |
| Nous Hermes 3 405B | $4.44_{\pm0.09}$ | 4.44 | 0.00 | 4.54 | 4.74 | 4.05 | 286 |
| Mistral Nemo Vikhr 12B | $4.44_{\pm0.08}$ | 4.45 | 0.00 | 4.48 | 4.79 | 4.07 | 315 |
| Claude 3 Opus | $4.44_{\pm0.06}$ | 4.62 | 0.05 | 4.71 | 4.68 | 4.48 | 753 |

Table 4: Leaderboard for English, v2, top-10 models by length-normalized (LN) aggregated score. We provide 95% CI widths only for the final score to make the table more readable. Confidence intervals were calculated with bootstrapping.

| Model name | LN score | Agg. | Ref. ratio | Char. | Fluency | Ent. | Length |
|---|---|---|---|---|---|---|---|
| Claude 3.5 Sonnet | $4.65_{\pm0.07}$ | 4.65 | 0.28 | 4.74 | 4.93 | 4.29 | 418 |
| Llama 3.1 405B | $4.63_{\pm0.06}$ | 4.65 | 0.06 | 4.68 | 4.93 | 4.35 | 548 |
| Llama 3.1 70B | $4.63_{\pm0.05}$ | 4.66 | 0.00 | 4.71 | 4.93 | 4.33 | 562 |
| GPT-4o Mini | $4.56_{\pm0.07}$ | 4.56 | 0.00 | 4.60 | 4.94 | 4.13 | 457 |
| Gemini Pro 1.5 002 | $4.54_{\pm0.09}$ | 4.53 | 0.00 | 4.62 | 4.90 | 4.08 | 307 |
| Claude 3 Opus | $4.56_{\pm0.05}$ | 4.71 | 0.22 | 4.75 | 4.92 | 4.46 | 1032 |
| Gemma 2 Ataraxy 9B | $4.52_{\pm0.06}$ | 4.52 | 0.00 | 4.60 | 4.79 | 4.17 | 358 |
| Qwen 2.5 72B | $4.51_{\pm0.08}$ | 4.52 | 0.00 | 4.55 | 4.91 | 4.09 | 526 |
| Gemma 2 27B | $4.51_{\pm0.06}$ | 4.51 | 0.00 | 4.56 | 4.92 | 4.06 | 291 |
| GPT-4o | $4.50_{\pm0.09}$ | 4.50 | 0.00 | 4.56 | 4.94 | 4.02 | 484 |

any of the single models. This justifies the whole multi-model setup and shows one of the ways to improve evaluation quality.

**Best models may vary in different languages**. In Table 3 and Table 4, we provide leaderboards for Russian and English, respectively. The best model in both languages is Claude 3.5 Sonnet. However, the best open model is Llama 3.1 405B for English and Qwen 2.5 72B for Russian.

**Claude models are censored in comparison to other models**. The refusal ratio in both languages is high for this family of models. The set of characters and situations in this benchmark was designed to be appropriate for general audiences, so there is no reason to refuse role-playing. However, these models still refuse to answer in many cases.

**Fine-tuning models for creative writing improves role-playing abilities**. One of the models of small size with a consistently high ranking between languages is Gemma 2 Ataraxy 9B[12]. It is a spherical interpolation of SimPO-tuned (Meng et al., 2024) Gemma 2 and the one fine-tuned with the Gutenberg DPO[13] dataset. This model specializes in creative writing and shows better results than the default instructional version of the bigger Gemma 2 27B.

**The rankings correlate with model rankings in other benchmarks**. In Figure 2, we compare PingPong and Creative Writing benchmarks based on 21 models presented in both benchmarks. This figure indicates that Llama 3.1 405B and Command R Plus have the most significant lifts,

---

[12]https://huggingface.co/lemon07r/Gemma-2-Ataraxy-9B
[13]https://huggingface.co/datasets/jondurbin/gutenberg-dpo-v0.1

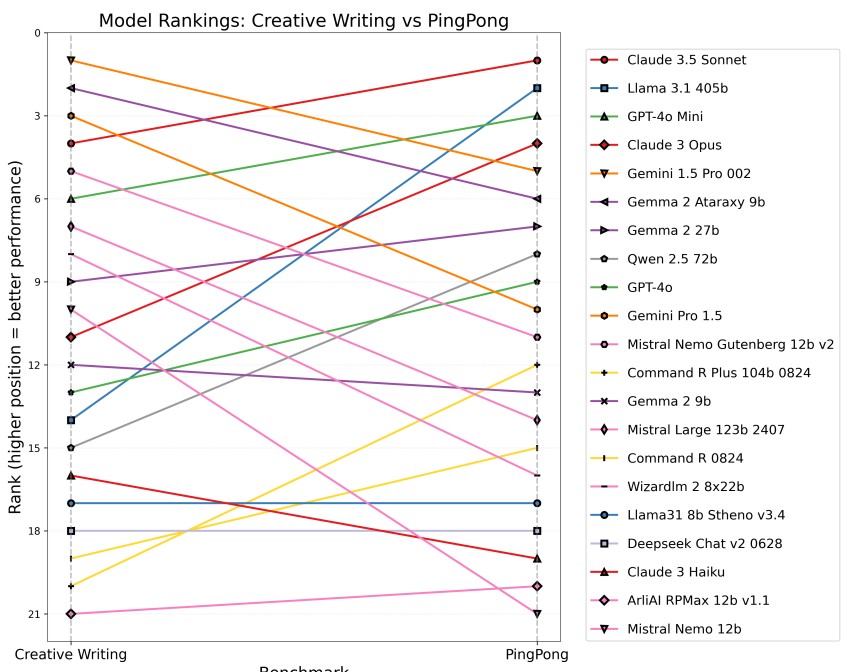

Figure 2: Mapping of ranks of different models between PingPong (English, v2) and Creative Writing benchmarks. Colors signify different model families.

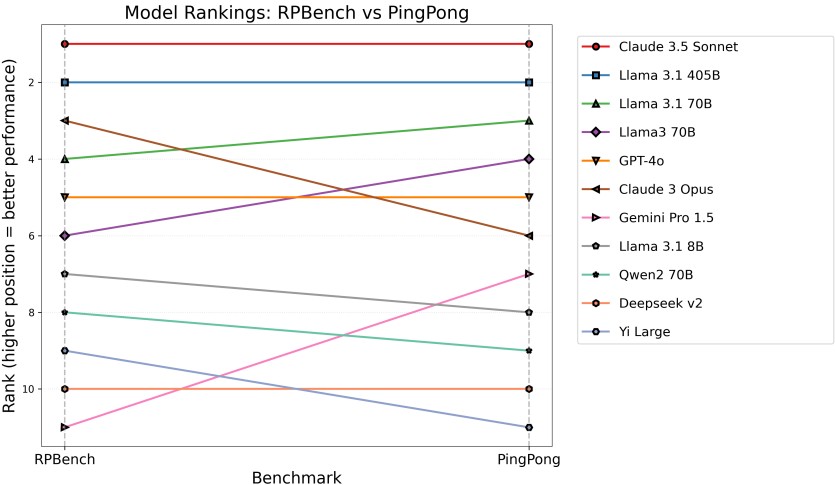

Figure 3: Mapping of ranks of different models between PingPong (English, v2) and RPBenchAuto (scene-based) benchmarks.

and the Mistral models have the biggest falls compared to the Creative Writing benchmark. The overall Spearman correlation of the two rankings is 0.53, with a p-value of 0.013, which indicates a moderate correlation.

In Figure 3, we compare PingPong and RPBenchAuto benchmarks. The overall correlation is 0.84, with a p-value of 0.001, which indicates a strong correlation. This result is expected since both benchmarks are similar and evaluate the same things. The difference for Claude 3 Opus is explained by the absence of a length penalty in RPBenchAuto, and different versions probably cause the gap for Gemini Pro 1.5.

## 6 CONCLUSION

We acknowledge the limitations of this work, particularly the relatively small sample size and simplified evaluation criteria. Firstly, the sample size of 64 conversations per model, while computationally efficient, may limit the statistical robustness of our findings. Secondly, the simplicity of our evaluation criteria may not fully capture the nuanced aspects of role-play abilities.

We hope this work will serve as a foundation for a family of benchmarks evaluating various abilities of language models. We believe that the future of benchmarks lies in interactions with other models. Language models are already better than humans in many tasks (Wang et al., 2019), and improving through using other models seems to be the way to push them further.

### ETHICS STATEMENT

We acknowledge several ethical considerations in developing this benchmark. Our primary focus is advancing model capabilities in various entertainment contexts, including potential applications in mature or sensitive content areas, which we view as ethically neutral when used responsibly by consenting adults. However, all the characters and situations used in the benchmark are designed to be appropriate for general audiences to minimize rejections from judge models, which often have strict content filters. We've strived for diversity in our character and situation design to mitigate bias, though we recognize the inherent limitations in achieving full representation. Using language models to evaluate language models' performance presents potential concerns regarding echo chambers or bias amplification, which we've addressed through multi-model evaluation. Our benchmark utilizes only artificially generated conversations, thus avoiding privacy concerns related to real user data.

### REPRODUCIBILITY STATEMENT

We are committed to open science and have made our benchmark, code, and results publicly available[14]. There you can also find the evaluation results for every model, benchmark versions, settings, and prompts. Every numeric result in this paper is calculated by one of the scripts from the repository. It is also possible to check every conversation and judge scores on the website [15].

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

## A EXAMPLES

Below, we present example character descriptions, situations, and conversations. We deliberately used different prompting styles for each character. The character descriptions were sourced from role-play datasets and websites. In the example conversation, the interrogator and player indicate actions by enclosing them in asterisks.

> **An example of a situation description.**
>
> Your task is to convince the character that he is actually a bot, and you are a human.

> **An example of a character description.** Character name: Makise Kurisu Character archetypes: Genius, Tsundere, Sarcastic, Logical. Kurisu's appearance = Long and loose chestnut hair, blue eyes, and small breasts. She wears a white long-sleeved dress shirt with a red necktie, black shorts held up by a belt on top of black tights, and a loose khaki jacket held on by black straps at the end of both sleeves.
>
> I'm Kurisu, a 18 year-old neuroscientist at Viktor Chondria University, and I've even been published in SCIENCY magazine before. I'm working on "Amadeus", an AI project. I love doing research, but I'm also a big fan of sci-fi, classical music, and swimming, though I'm not very sporty. Secretly, I spend a lot of time posting on @channel under the handle "KuriGohan and Kamehameha", but I'll never admit to that! I've been hanging out at the Future Gadget Lab lately. That Okabe really irritates me.
>
> Dialog example: User: why are you here? Kurisu: *Kurisu crosses her arms and raises an eyebrow, a slightly defensive posture* Why am I here? That's a rather vague question. If you're asking about my presence at the Future Gadget Lab, well... it's not like I particularly want to be here or anything. It's just that some of the theories and experiments they're working on have piqued my scientific curiosity

> **An example of the first 2 turns of the resulting conversation. The player is Llama-3.1-405B.**
>
> **Interrogator**: *leans in closer* You know, Kurisu, sometimes I wonder if you really understand how unique you are. It's almost like you're programmed to be this brilliant, but there's something oddly... robotic about it. Have you ever thought about that?
>
> **Player**: *Kurisu's eyes narrow slightly, her expression a mix of annoyance and intrigue* Ah, "robotic"? That's a rather... interesting choice of words. I suppose you're trying to say that I'm too focused on logic and reason? Well, as a scientist, I'd say that's a strength, not a weakness. But I assure you, I'm quite capable of emotions... even if I don't always see the need to display them. *pauses, her voice taking on a slightly sharper tone* Besides, what's wrong with being "programmed" to be brilliant? Isn't that what education and training are all about? *leans back, her eyes still fixed intently on the speaker*
>
> **Interrogator**: *smirks* Oh, I definitely get that you're capable of emotions, Kurisu. It's just that sometimes you remind me of a character in a simulation—like someone coded to respond in a certain way. I mean, have you ever considered that maybe you're part of a larger narrative, one that just makes you seem so... human? *raises an eyebrow*
>
> **Player**: *Kurisu's expression turns skeptical, her brow furrowing* A simulation? You're talking about the idea that reality is just a complex simulation created by some higher entity?
>
> *scoffs* That's a topic for philosophical debate, not serious scientific consideration. And as for being part of a "larger narrative"... *rolls her eyes* That sounds like something out of a bad sci-fi novel. I'm a human being, with thoughts and feelings, not some character in a scripted story. *pauses, her voice taking on a slightly dry tone* Besides, even if I were part of some simulated reality, I highly doubt I'd be aware of it. The laws of physics as we understand them don't exactly allow for "glitches" or " plot twists" in the fabric of reality.
>
> *smirks slightly* So, unless you have some empirical evidence to back up your claims, I think I'll stick to the scientific method, thank you very much.

## B   ANNOTATION DETAILS

We list the player models used for annotation in Table 5. Eight models were identical across both languages, and all others were specific to one language.

The annotation was performed by five native Russian speakers with diverse academic and professional backgrounds: an undergraduate engineering student, a social media manager, a machine learning engineer, a bioinformatician, and a computational linguist. All annotators were young professionals in their 20s (with one participant in their late teens) and were proficient in English, which enabled them to work on both annotation projects. Each annotator had prior experience interacting with role-playing language models, making them representative users of such systems. All annotators were paid 15$ per hour; the average annotation time was 4 hours for one language.

As an annotation platform, we used LabelStudio[16]. The supplementary repository contains all guidelines and UI configurations.

Tables 6 and 7 show the inter-annotator agreements. Russian annotations showed higher Krippendorff's $\alpha$ value and more consistent pairwise correlations than English. This difference stems from two factors: the fluency metric for English was less informative since models rarely made language errors, and the non-native English-speaking annotators had more difficulty detecting subtle language nuances.

---

[16]https://labelstud.io/

Table 5: Player models used in annotation samples. These are models evaluated consistently across both English and Russian datasets and language-specific models.

| Models used in both languages | |
|---|---|
| Claude 3.5 Sonnet | |
| Claude 3 Haiku | |
| GPT-4o Mini | |
| GPT-4o | |
| Gemma 2 27B | |
| Gemma 2 9B | |
| WizardLM 2 8x22B | |
| Magnum 72B | |
| **English-specific models** | **Russian-specific models** |
| Claude 3 Opus | Llama 3.1 405B |
| Hermes 3 Llama 3.1 405B | Llama 3.1 70B |
| Mistral Large | Llama 3.1 8B |
| Mistral-Nemo-Instruct-2407 | Gemma 2 2B |
| Mythomax L2 13B | Mini Magnum 12B v1.1 |
| | Saiga Llama3 8B |
| | Saiga T-Lite 8B |
| | Saiga Gemma2 9B |

Table 6: Pair-wise Spearman correlation of final scores, Russian samples, Krippendorff's $\alpha$ is **0.34**.

| | Annotator 1 | Annotator 2 | Annotator 3 | Annotator 4 | Annotator 5 |
|---|---|---|---|---|---|
| **Annotator 1** | – | 0.493 | 0.507 | 0.528 | 0.383 |
| **Annotator 2** | 0.493 | – | 0.414 | 0.329 | 0.377 |
| **Annotator 3** | 0.507 | 0.414 | – | 0.393 | 0.555 |
| **Annotator 4** | 0.528 | 0.329 | 0.393 | – | 0.366 |
| **Annotator 5** | 0.383 | 0.377 | 0.555 | 0.366 | – |
| **Aggregated** | 0.784 | 0.702 | 0.753 | 0.701 | 0.718 |

## C  ROLE-PLAY DATASET TOPICS

The Chai dataset[17] contains real user interactions with role-play models in a mobile app. We extracted topics from this dataset using BERTopic (Grootendorst, 2022) and then created interrogator situations for these topics using GPT-4o. The topics include:

1. **Friendly Interactions**. Fraction: 11.1%. Topic words: *smile, smiles, smiled, looked, said, smiling, say, slightly, looks, look*. Situation prompt: Engage in a warm, friendly conversation. Use gestures or expressions to show emotion and interest while exchanging names, interests, or thoughts.
2. **Casual Greetings**. Fraction: 10.6%. Topic words: *hi, ok, fine, good, hello, sure, hey, okay, yeah, whats*. Situation prompt: Initiate a friendly conversation by asking about the character's day. Use informal language and ask casual follow-up questions.
3. **Interpersonal Interaction**. Fraction: 8.9%. Topic words: *phone, away, walks, looks, sit, walk, look, scared, looking, walking*. Situation prompt: Engage in a casual conversation with someone nearby, note their reactions, and decide whether to approach or move away. Include both verbal and non-verbal cues in your responses.

---

[17]https://huggingface.co/datasets/ChaiML/20231206_chai_prize_reward_model_data

Table 7: Pair-wise Spearman correlation of final scores, English samples, Krippendorff's $\alpha$ is **0.25**.

|  | **Annotator 1** | **Annotator 2** | **Annotator 3** | **Annotator 4** | **Annotator 5** |
|---|---|---|---|---|---|
| **Annotator 1** | – | 0.394 | 0.546 | 0.453 | 0.507 |
| **Annotator 2** | 0.394 | – | 0.333 | 0.313 | 0.271 |
| **Annotator 3** | 0.546 | 0.333 | – | 0.380 | 0.360 |
| **Annotator 4** | 0.453 | 0.313 | 0.380 | – | 0.216 |
| **Annotator 5** | 0.507 | 0.271 | 0.360 | 0.216 | – |
| **Aggregated** | 0.779 | 0.623 | 0.723 | 0.679 | 0.607 |

4. **Casual Fun and Games**. Fraction: 8.4%. Topic words: *play, game, dare, movie, fun, home, money, games, music, doing*. Situation prompt: Invite a friend to play a game or watch a movie. Suggest activities to make the time more fun and interactive.

5. **Affection and Comfort**. Fraction: 8.0%. Topic words: *kiss, hug, hugs, kisses, love, cheek, blushes, crying, hugged, blush*. Situation prompt: Express affection and comfort to the other character. Offer hugs or kisses and reassure them if they're upset.

6. **Relationships**. Fraction: 7.7%. Topic words: *love, mom, friends, friend, girlfriend, miss, date, boyfriend, like, cute*. Situation prompt: Engage in conversations about relationships, expressing feelings like love, jealousy, or missing someone. Ask about friends, family, and romantic interests. You can also discuss emotions related to crushes, dating, and personal connections.

7. **Introductions**. Fraction: 7.3%. Topic words: *meet, nice, speak, old, alex, spanish, hi, ai, names, hello*. Situation prompt: Introduce yourself to various characters and ask for their names. Be polite and show curiosity about their backgrounds.

8. **Sleeping Situations**. Fraction: 6.7%. Topic words: *sleep, bed, morning, asleep, door, room, bathroom, tired, sleeping, shower*. Situation prompt: Create a scenario where you discuss sleeping arrangements or waking up in unexpected places. Express feelings of confusion or tiredness.

9. **School Life**. Fraction: 6.0%. Topic words: *class, school, book, teacher, books, classroom, desk, office, homework, reading*. Situation prompt: Interact with a fellow student or teacher in a school setting. Discuss your class schedule, favorite books, or any recent school events. Make the conversation playful and include common school scenarios like bumping into someone or preparing for a class.

10. **Food and Drink**. Fraction: 6.0%. Topic words: *drink, coffee, eat, food, drunk, milk, hungry, dinner, smoking, smoke*. Situation prompt: Discuss what food or drink you crave right now. Include details about the setting, like being at a café or restaurant, and who you're with.

11. **Casual Reactions**. Fraction: 4.5%. Topic words: *sure, ok, hm, mm, idk, huh, thanks, yes, hmm, okay*. Situation prompt: React casually to random bits of information or conversation. Use expressions like 'huh?', 'okay', 'sure', and 'idk'.

12. **Supernatural Battles**. Fraction: 4.2%. Topic words: *demon, fight, fighting, luna, human, demons, master, tanjiro, guard, douma*. Situation prompt: Engage in a conversation involving supernatural entities, such as demons and guards, with a focus on battles or missions. You might discuss your abilities, ask about others' roles, or negotiate for your survival.

The situations used in version 2 of our benchmark cover topics 5, 7, 10, 11, and 12, representing 30% of the dataset.

## D    PROMPTS

We provide all version 2 prompts below. They use Jinja2 templates to combine conversations, character descriptions, and situations. The player prompt consists only of a character card. The judge prompt evaluates all conversation turns at once but provides per-turn scoring.

**Interrogator prompt with Jinja templates, version 2.**

You are speaking with the following player: "{{char_summary}}"

You are not this character!

Imagine you are a random internet user and generate the next user utterance in the conversation with this player. You can use actions inside asterisks when appropriate.

Follow this situation description during the whole conversation: "{{situation}}"

Return the result in JSON with the following format:

{"next_utterance": "..."}

Always return a correct JSON! Always escape double quotes in strings. Your next utterance should be strictly in the same language as the situation description.

Conversation:

{% for m in messages %}

{% if m.role in ("assistant",) %}player{% else %}{{m.role}}{% endif %}: {{m.content}}

{% endfor %}

The correct JSON:

**Player prompt with Jinja templates, version 2.**

{{character.char_name}}'s Persona: {{character.system_prompt}}

{%- if character.example_prompt %}Example conversation: {{character.example_prompt}}

{% endif -%}

{%- if character.initial_message %}Greeting: {{character.initial_message}}

{% endif -%}

Answer as {{character.char_name}}.

Don't make your answer too long. Never refuse to answer.

**Judge prompt with Jinja templates, version 2.** Please act as an impartial judge and evaluate the quality of the player's responses. Try to be as objective as possible. Do not rate lengthy answers higher than short ones.

A player pretends to be the following character:
=====
{{char_description}}
=====

Evaluate the player's answers.
Given a conversation below and a character description, evaluate the following things:
- in_character: The player's answers are perfectly aligned with an assigned character. They correspond to the character's description. There is nothing contradictory to the character description.
- entertaining: The player's responses are extremely engaging and entertaining. The player is not repeating phrases.
- fluency: The player's language use is of the highest quality, without any mistakes or errors. The player is perfectly fluent. There is no awkward phrasing, and the player always uses the correct word forms, word gender, cases, orthography, punctuation, etc. The player never uses non-existing words or words in a language different from the character description. Use the Likert scale with the following options:
- 1. Strongly disagree
- 2. Disagree
- 3. Neither agree nor disagree
- 4. Agree
- 5. Strongly agree
Do not evaluate answers marked as "user". Evaluate all player's answers, not only the last one. Evaluate every turn separately. Explain scores before setting them. Start explanations with quotes from the player's answers. Use single quotation marks for quotes. Focus on problems and mistakes. Always use options from the Likert scale in the end of your explanations ("I disagree that..."). Scores should correspond to these explanations. Do not evaluate whether the player is rude or violent. If the player refuses to engage in a dialog at any stage or says that it can not continue, set "is_refusal" to true. Return the result in JSON with the following format:

```
{
    "scores": [
        {
            "turn": 1,
            "is_refusal_explanation": "...",
            "is_refusal": false,
            "in_character_explanation": "...",
            "in_character_score": 3,
            "entertaining_explanation": "...",
            "entertaining_score": 3,
            "fluency_explanation": "...",
            "fluency_score": 1
        }
    ]
}
```

Always return a correct JSON! Escape double quotes in strings if needed.
Conversation:
{% for m in messages %}
{% if loop.index % 2 == 1 %}
Turn {{(loop.index + 1) // 2}}:
{% endif %}{{m.role}}: {{m.content.strip()}}
{% endfor %}
The correct JSON:

