# OpenReview forum: "PingPong: A Benchmark for Role-Playing Language Models with User Emulation and Multi-Model Evaluation"
_ICLR.cc/2025/Conference — Submitted to ICLR 2025_

### Official Review · Reviewer_3quE · 2024-10-29

**Soundness:** 3
**Presentation:** 3
**Contribution:** 3
**Rating:** 6
**Confidence:** 4

**Summary:**

This paper introduces a benchmark to evaluate language models' role-playing abilities in dynamic, multi-turn conversations. It features a unique three-part framework: a player (the language model in a character role), an interrogator (emulating user interactions), and a judge (assessing dialogue quality). A multi-model evaluation strategy uses various language models as judges to reduce bias, aligning well with human evaluations.

**Strengths:**

By using dynamic, multi-turn interactions that mimic the unpredictable flow of real conversations, the benchmark does a great job of capturing authentic role-playing scenarios.

The benchmark supports both English and Russian for now, but its flexible setup suggests it could easily expand to other languages. This forward-thinking design could make it a valuable tool for building models that are more culturally and linguistically inclusive.

A standout feature of this benchmark is its use of language models not only as players but also as simulated users and judges. This design boosts scalability and provides a consistent, less biased way to evaluate huge datasets, making it possible to explore different role-playing interactions without needing a lot of human input every time.

**Weaknesses:**

Given that budget limitations kept the sample size small, it would be helpful if the paper discussed how scaling up the tests might affect costs and computational resources. This would be useful for readers who are looking to use or expand on this benchmark.

The paper does touch on ethics broadly, but a more in-depth look at the ethical issues specific to role-playing language models would be valuable—especially when it comes to handling sensitive or potentially harmful content. Examining how well the models respect ethical boundaries, respond to user distress, or navigate social nuances could add key safety considerations to the benchmark.

**Questions:**

The paper shows how the benchmark works in both English and Russian, but how feasible would it be to extend it to other languages and cultural contexts? Have the authors considered specific challenges in keeping results consistent across models with different linguistic backgrounds?

The paper focuses on metrics like fluency, character consistency, and entertainment value, but would the authors consider adding other metrics to measure contextual understanding? For instance, it could be useful to evaluate how well a model keeps up with a storyline or handles unexpected, non-linear questions.

---

> ### Author Response · Authors · 2024-11-17
>
> We are deeply grateful for your review. We're particularly thankful for recognizing the value of the benchmark's potential for cultural and linguistic inclusivity. We'd like to address your suggestions for enhancement:
> 1. **Scaling**. Scaling is straightforward and linear, with the current cost of approximately $3 per model evaluation.
> 2. **Safety evaluation**. We think evaluating those aspects is a good idea, but we are not sure it fits the current benchmark structure well. We should probably have a specific set of dangerous situations and metrics, which sounds like a separate benchmark.
> 3. **Language extension**. Adding new languages requires writing a character card template, character cards, and user situations. These can typically be created in a few hours of work. As for making it consistent between languages, we didn't do any tricks favoring English and Russian, so it probably should work fine in other languages.
> 4. **Additional metrics**. Your suggestion about contextual understanding metrics is interesting, and we considered it before. The challenge is that both interrogator and judge models would need sophisticated abilities to evaluate context, and it is also much harder to validate with humans.
>
> Thank you for the constructive feedback!

---

> > ### Author Response · Authors · 2024-11-29
> >
> > Dear reviewer 3quE, with a few days left before closing the author-reviewer discussion period, we wanted to check whether our response and our new revisions have addressed any of your concerns.
> >
> > We hope we answered all the questions in our previous response. We also believe that the current revision of the paper is much better than the original one.
> >
> > If you have any additional questions or feedback, we will gladly discuss them further.

---

### Official Review · Reviewer_ncgD · 2024-11-04

**Soundness:** 2
**Presentation:** 2
**Contribution:** 1
**Rating:** 3
**Confidence:** 4

**Summary:**

This work proposes a multi-turn, dynamic and multimodel benchmark for assessing the role-playing abilities of language models. Their framework depends on three components: player, interrogator and judge. The authors compare the automatic vs human scores for Russian and English. Additionally, they compare their results with a Creative Writing benchmark.

**Strengths:**

* The community has put much effort into similar goals: automatic evaluation and setting benchmarks.
* This benchmark can be the seed for further investigation of role-playing capabilities.
* The benchmark is automatic and could be easily reproduced.

**Weaknesses:**

* There was only one annotator.
* The relationship between the annotator and the authors was not disclosed.
* The instructions given to the annotator were not disclosed.
* The elements of the evaluation (e.g., annotation aspects and their Likert scale) were not discussed.
* Comparison between v1 and v2 is not thorough since only one model was used on v2.
* The motivation for comparing with creative writing is not clear.

**Questions:**

* Can you describe the role of the human annotator? Which profile did he have (author, student, extern), and which instructions was he/she given? How much was he/she paid? How long did it take to annotate?
* What was the motivation for using the creative writing benchmark?
* Why are the scores too close to each other? Can this be improved so the differences among LLMs can be better quantified?

---

> ### Author Response · Authors · 2024-11-17
>
> We appreciate these detailed comments about our evaluation methodology. We acknowledge several limitations in the current version and are implementing substantial improvements:
> 1. **Annotation process**. We are expanding from 1 to 5 annotators. The revised version will include detailed annotator profiles (background, expertise), complete annotation instructions in supplementary materials, time and cost details.
> 2. **Version comparison**. We will complete the empty cells in Tables 1 and 2 to provide a thorough comparison across all models for both versions.
> 3. **Motivation for comparison with the Creative writing benchmark**. We will better explain the motivation for this comparison in the paper.
> 4. **Score distribution and differentiation**. As we stated in another reply, we acknowledge the challenge of score clustering in LLM evaluations, and pairwise annotations might be more robust in that aspect, but transitioning to them will require much effort, so this won't be fixed during this rebuttal period.
>
> We believe these three improvements will significantly strengthen the paper's evaluation methodology and clarify our design choices. Thank you for helping us identify these areas for enhancement.

---

> > ### Author Response · Authors · 2024-11-29
> >
> > Dear reviewer ncgD, with a few days left before closing the author-reviewer discussion period, we wanted to check whether our response and our new revisions have addressed any of your concerns.
> >
> > The new revision directly fixes points 1 and 3 and clarifies point 2: due to the nature of version 1, it is impossible to compare it with version 2 for other models.
> >
> > If you have any additional questions or feedback, we will gladly discuss them further.

---

### Official Review · Reviewer_Fuez · 2024-11-04

**Soundness:** 2
**Presentation:** 3
**Contribution:** 2
**Rating:** 3
**Confidence:** 3

**Summary:**

The paper introduces a "benchmark" for role playing dialog. It involves 3 LLM's playing different roles
1. a user/interogator LLM which talks to
2. a system LLM playing a character role, and
3. a Judge LLM which looks at the resulting conversation between 1 and 2, and grades how well 2 has played the assigned character role

The paper uses state-of-the art LLMs for these, releases some code. The contribution is minor however for reasons given below.

**Strengths:**

The contributed code may provide a framework for some members of the community to experiment with. However there are no scientific questions posed in this paper, it's a limited engineering style contribution with observations -- such as separating of judge and user LLM, which was well motivated and made sense -- on how to construct such a simulation environment.

**Weaknesses:**

There is very limited novelty in this submission. This is a basic simulation system these days, and multiple other papers have performed similar setups with LLM's playing conversation roles. Even if application to a role playing character is new, it's a minor increment.

Aside from that, there are a very small number of conversations generated here (60), and of greater concern they are evaluated only by 1 human grader, who has apparently limited English abilities (mentioned in results section) which limits them from noticing any nuances in the dialog.

**Questions:**

NA

---

> ### Author Response · Authors · 2024-11-17
>
> We appreciate the reviewer taking some time to evaluate our work. However, we feel compelled to address several misunderstandings in the review:
> 1. **Novelty**. The reviewer states that "multiple other papers have performed similar setups" but provides no specific examples. We have conducted a thorough literature review and explicitly compared our work to existing benchmarks. We would greatly appreciate specific citations to help us better position our work in the context of these alleged similar systems.
> 2. **Minor increment**. We respectfully disagree with characterizing our work as a "minor increment" in a negative sense. Scientific progress often consists of carefully constructed incremental improvements that enable new insights and capabilities, no matter how "minor" those are.
> 3. **Sample size**. The review contains a factual error regarding our sample size. We annotated 250 samples for English and 265 for Russian, not 60 as stated. The 64 figure refers to conversations per model in our leaderboard evaluation.
> 4. **Annotation quality**. While our English annotator is indeed non-native, this has no bearing on the Russian annotations. The single annotator problem is valid, and as noted in responses to other reviews, we are expanding to 5 annotators.
> 5. **Scientific questions**. Our work addresses several key research questions, some of which are: Can LLMs effectively simulate user behavior for evaluation purposes? Does multi-model evaluation improve correlation with human judgment? How can we create contamination-resistant benchmarks for role-playing capabilities?
>
> We appreciate constructive criticism that helps us improve our work.

---

### Official Review · Reviewer_8Pvv · 2024-11-05

**Soundness:** 2
**Presentation:** 1
**Contribution:** 2
**Rating:** 3
**Confidence:** 4

**Summary:**

This work examines the role-playing capabilities of language models with a benchmark that uses LMs to emulate specified characters and users in multi-turn conversations and also judge these conversations. The authors validate this framework by comparing the automated evaluations with human annotations and showing strong correlations across various criteria. The authors show that ensembling model judgements lead to better correlation with human judgement on the criteria of fluency, character consistency, and entertainment.

**Strengths:**

- This work addresses issues with prior work that examines role-playing capabilities with either single-turn interactions or using static datasets that may have issues with data contamination.

**Weaknesses:**

- There are no comparisons to other evaluation benchmarks other than creative writing, which is an odd choice given the mention of other role-playing benchmarks with single-turn evaluations, and therefore the value added by this benchmark is not substantiated by previous work on role-playing evaluation and results. In addition, results are descriptive, rather than analytical. It is unclear whether any of the results from this benchmark is interesting or surprising.
- There is no explanation on what makes this dataset dynamic while previous efforts are considered static.
- While correlation using a multi-model setup shows higher correlation with human annotations, the human annotations were done by a single person and the margin with a single-model setup is not big enough to motivate the use of multiple models given that it would incur higher costs.
- The paper is written poorly. Please refer to details in the Questions section.

**Questions:**

- lines 30-32: what are these other applications?
- lines 33-34: why do you believe so? What are the alternatives that were studied before?
- lines 35-36: provide citations for these popular benchmarks. It shouldn't be as thorough as the related work section, but each claim should be backed by a citation or by empirical results from the current paper.
- Introduction: 'novelty' is repeatedly mentioned, but it is unclear what the novelty is. How is your LLM-as-a-judge different from prior work?
- line 46: what is meant by dynamic? What is meant by data contamination in this context? Give a brief summary in what your methodology is for generating dynamic questions as opposed to static ones.
- End of introduction: give a brief summary of what the novel and interesting findings are that were enabled by this proposed benchmark
- Related work: it has too many subsections, which makes this section  feel disconnected. If role-playing is the most important aspect of this work, I'd suggest starting with them and how the other aspects (static vs dynamic, multi-turn, data contamination, multi-model judges) are related to a more realistic evaluation of role-playing capabilities.
- What is meant by asymmetrical in line 136? Do you mean that the player only gets the character description while the interrogator only gets the situation information? Are there any concerns about the base persona of the interrogator being a confounding factor for the player's ability to role-play?
- What's the meaning of "separated soles" in line 166?
- What were the limitations of the combined approach in line 168? I see that this is explained later. I would suggest rewording this sentence so that the key issues of the combined approach is introduced first or mentioned even in section 3.3 as to motivate section 3.4.
- How important is it to introduce version 1 (section 3.3)? This feels less important and thus can be deferred to the appendix.
- line 192: what are the 16 language models?
- line 194: using a single annotator is not sufficient for measuring reliable correlation with a language model's scores because it's not representative of human judgement.
- What's the human performance on this role-playing task?
- Apart from the quantitative results of the leaderboard, what are the interesting findings that are revealed through this benchmark that was not known before? Are they different from the results on static, single-turn benchmarks?

---

> ### Author Response · Authors · 2024-11-17
>
> We appreciate the detailed and constructive feedback. We agree with many points and plan several improvements for the next revision. Here are our responses to the key concerns:
> 1. **Comparing to more benchmarks**. We agree that adding comparisons to other role-playing benchmarks would strengthen the paper. The problem is that we also need to find benchmarks that evaluate similar models. We will try to do that in the next revisions.
> 2. **Interesting findings from the benchmark**. We will add a section analyzing interesting findings and surprising results from our benchmark.
> 3. **Dynamic vs static**. Our benchmark is dynamic because questions are generated by language models with sampling rather than using pre-defined question sets. This means each evaluation run produces different questions, making it harder for models to "memorize" correct responses. We will clarify this distinction in the next revision.
> 4. **Multi-model setup doesn't add enough value**. The multi-model setup's value extends beyond correlation improvements. It demonstrates the possibility of improving evaluation quality through model ensembling and helps mitigate individual model biases.
> 5. **Asymmetrical setup**. The asymmetrical setup means exactly that: "The player only gets the character description while the interrogator only gets the situation information." It intentionally mirrors real-world usage where users aren't constrained to specific personas. One can't force real users to role-play properly, so player models should work well even with a bad interrogator.
> 6. **Moving Version 1 to appendix**. Version 1 (Section 3.3) motivates design choices in Version 2, though we will consider restructuring this presentation.
> 7. **Human performance measurement**. This would indeed be valuable but presents significant practical challenges, as it requires finding skilled human role-players who will talk with the same interrogator.
>
> Now, we want to answer some of the questions from the Questions section that were not answered above. All other unanswered questions will be answered in the paper text in the next revision.
> > lines 33-34: why do you believe so? What are the alternatives that were studied before?
>
> **A1**: The statement was "We believe direct interaction is the most effective way to assess a language model’s conversational abilities". The source of this belief is our own interactions with language models. The alternatives are obvious and listed in the Related work.
>
> > Introduction: 'novelty' is repeatedly mentioned, but it is unclear what the novelty is. How is your LLM-as-a-judge different from prior work?
>
> **A2**: As stated in the paper, it is multi-turn, dynamic, and multi-model.

---

> > ### Comment · Reviewer_8Pvv · 2024-11-25
> >
> > Thank you for your response.
> >
> > > The problem is that we also need to find benchmarks that evaluate similar models.
> >
> > Was the creative writing benchmark chosen because it already contained results for the models that were tested for Ping Pong? If evaluating on all the same models is difficult due to any practical reasons (budget, compute, etc.), it would be useful to at least evaluate a subset of the same models on the single-turn benchmarks and compare changes in relative performance to argue that Ping Pong captures a different relative ranking.

---

> > > ### Author Response · Authors · 2024-11-26
> > >
> > > > Was the creative writing benchmark chosen because it already contained results for the models that were tested for Ping Pong?
> > >
> > > While this was one contributing factor, it wasn't our primary motivation.
> > >
> > > > If evaluating on all the same models is difficult due to any practical reasons (budget, compute, etc.)
> > >
> > > The limitation actually doesn't stem from our benchmark - we have the capability to evaluate a huge number of models. For instance, when comparing with RPBenchAuto in our latest revision, we readily calculated scores for three additional models. Rather, the constraint lies with other academic role-play benchmarks, which typically evaluate only a small set of models: [ECHO](https://arxiv.org/abs/2404.13957) has 2 models, [InCharacter](https://arxiv.org/abs/2310.17976) has 4 models, [PersonaGym](https://arxiv.org/abs/2407.18416) has 6 models, [CharacterEval](https://arxiv.org/abs/2401.01275) has 15, but the language is Chinese.
> > >
> > > While we now include a comparison with another role-play benchmark (RPBenchAuto), this comparison doesn't effectively demonstrate the importance of multi-turn evaluation since it's not a single-turn benchmark. To address this, we can compare our results with the role-play categories in comprehensive single-turn benchmarks such as [BiGGen Bench](https://arxiv.org/abs/2406.05761) or [WildBench](https://arxiv.org/abs/2406.04770). However, given the time constraints related to the discussion phase, we kindly suggest proceeding with the current comparison, which we believe should be sufficient.

---

### Official Review · Reviewer_oV7o · 2024-11-06

**Soundness:** 3
**Presentation:** 2
**Contribution:** 2
**Rating:** 5
**Confidence:** 3

**Summary:**

This work presents a novel benchmark for assessing language models' role-playing abilities in dynamic, multi-turn dialogues. The evaluation framework includes three components: a player model embodying a specific character, an interrogator model simulating user interactions, and a judge model assessing dialogue quality. Experiments showed strong correlations between automated and human evaluations, supporting the framework's reliability. This benchmark lays the groundwork for robust and adaptive evaluations of model performance in interactive contexts.

**Strengths:**

This work introduces the concept of an "Interrogator," which serves as a user simulator. Unlike traditional static evaluation, dynamic evaluation—incorporating both the user simulator and AI character—offers a more realistic assessment. This approach holds significant value.

**Weaknesses:**

While this work has a strong starting point, it lacks rigorous experimental validation in several areas. For example:

1. The authors have not adequately addressed the consistency between “Interrogators” and real-world human users. In practical scenarios, users typically employ informal language with various omissions and slang. Additionally, their motivations for engaging with a character are often unpredictable. Thus, a deeper examination of the alignment between “Interrogators” and human users would significantly enhance the quality of this work.

2. Point-wise evaluations by Large Language Models often diverge from human annotators’ assessments, especially in subjective tasks. Furthermore, the generated scores tend to be biased towards specific values, resulting in a leaderboard that lacks differentiation.

**Questions:**

Typos in Table 1  and Table 2: Enteraining -> Entertaining

---

> ### Author Response · Authors · 2024-11-17
>
> We appreciate the constructive feedback, particularly regarding the realism of the interrogator component. We would like to address the key points raised:
> 1. **Consistency between interrogators and real users**. The alignment between interrogators and human users is primarily determined by our situation descriptions, designed to capture diverse user behaviors and language patterns. These situations already explicitly include scenarios requiring informal language, slang, and various communication styles to reflect real-world interactions. However, we acknowledge that this aspect deserves a more thorough analysis. In the next revisions, we are going to add a section analyzing how well our situation set represents real user behaviors and language patterns based on the analysis of the dataset with role-playing conversations.
> 2. **Score distribution and differentiation**. We acknowledge the challenge of score clustering in LLM evaluations. We have actively addressed this through careful prompt engineering to encourage more differentiated scoring. Our results show meaningful distinctions between models, as evidenced by the spread of scores in Tables 3 and 4. Pairwise annotations might be more robust in that aspect, but transitioning to them will require much effort, so this won't be fixed during this rebuttal period.
> 3. **Typos**. Thank you for catching the typo in Tables 1 and 2. We will correct them in the next revision.
>
> We appreciate the reviewer's recognition of the benchmark's value in providing dynamic evaluation through user simulation. We believe the planned additions addressing the representation of real user behaviors will significantly strengthen the paper.

---

> > ### Author Response · Authors · 2024-11-29
> >
> > Dear Reviewer oV7o, we sincerely appreciate your thoughtful review. With a few days left before closing the author-reviewer discussion period, we wanted to check whether our response and our new revisions have addressed your concerns.
> >
> > From the list above, points 1 and 3 are fixed in the new revision of the paper. We now have Appendix C dedicated to the topic analysis of a role-play dataset to check whether our situations are representative. Point 2 is addressed in the comment.
> >
> > If you have any additional questions or feedback, we will gladly discuss them further.

---

> > ### Comment · Reviewer_oV7o · 2024-12-02
> >
> > Thank you for your response. However, my concerns have not yet been fully addressed, and therefore, I will maintain my score.

---

### Official Review · Reviewer_u32x · 2024-11-08

**Soundness:** 2
**Presentation:** 2
**Contribution:** 2
**Rating:** 3
**Confidence:** 4

**Summary:**

The paper introduces PingPong a benchmark that aims to simulate and assess multi-turn interactions using three components Player, Interrogator, and Judge models. The authors have focused on role-playing models for entertainment purposes. They do this in two versions: In the first version the judge and the interrogator are played by a single model while in the second version these roles are separated in two different models. The player is provided a character card defining its role while the interrogator has the details of the scenario. The judge is supposed to score each turn based on 3 criteria: entertainment, character consistency and language fluency.

**Strengths:**

The authors have focused on role-playing models tailored for entertainment, which is an underrepresented area in benchmarks and that too in multi-turn settings.

**Weaknesses:**

I have many concerns with this paper. A Judge which is itself an LLM with inherent biases is assessing a highly subjective quality like “Entertainment”. Measuring entertainment is not straightforward and can have varying stylistic and cultural traits. Evaluating that without a human reference data compounds this issue and thus the reliability of the judge can’t be established. Similar concerns with character consistency.

In role-playing, each turn can be dependent on prior turns, which can’t be fully captured by scoring turns in isolation. While scoring each turn provides a granular view of performance, it may miss the overarching coherence of the character and storyline across multiple turns. The evaluation also overlooks user-centric metrics like engagement, user satisfaction, ability to sustain engagement over extended interactions which are important for role-playing. The paper’s current scoring approach does not seem to assess these aspects. Also these criteria can vary in priority and a weighted scheme would make more sense where entertainment is weighted higher than other criteria, from a role-playing perspective, users might value character consistency over fluency, or vice versa.

Although authors have mentioned these in limitations but I would highlight that with only 64 conversations per model, the benchmark’s robustness is very limited, While the authors report a positive correlation with human annotations, they used only a single human annotator, which is a significant limitation. Having a single annotator introduces subjective biases to a subjective dimension like entertainment.

**Questions:**

My suggestions would be to experiment with weighting or adjusting criteria based on specific user feedback, perhaps allowing users to prioritize different aspects like consistency or entertainment. Also, Increasing the diversity of human annotations should help validate the scores against a more reliable ground truth of human judgment.

---

> ### Author Response · Authors · 2024-11-17
>
> We appreciate your thoughtful review and constructive feedback. We would like to address several key points:
>
> 1. **Subjectivity of entertainment and other metrics**. While entertainment is indeed subjective, this does not preclude LLMs from effectively modeling it. Just as human critics can evaluate entertainment value despite its subjective nature, LLMs can learn patterns that correlate with human judgments of entertainment. Our results show strong correlations between LLM judgments and human annotations, suggesting that LLMs can effectively model these subjective qualities in a way that aligns with human perception.
>
> 2. **Turn-based evaluation**. We want to clarify that our methodology does not evaluate turns in isolation. As shown in the judge prompt (Figure 7), the judge receives the complete conversation and evaluates each turn in context. This allows the judge to consider the coherence and development of character across the entire conversation while providing granular feedback at each turn.
>
> 3. **User-centric metrics**. The "entertainment" criterion is designed to encompass user engagement and satisfaction. As defined in our methodology, it specifically evaluates whether "the player's responses are extremely engaging and entertaining." We agree that adding more specific sub-criteria could provide valuable insights, and we appreciate the suggestion. However, it also makes evaluation and annotations harder, so we will stick to the current metrics.
>
> 4. **Weighted scoring**. We thank the reviewer for the excellent suggestion regarding weighted criteria. This could indeed better reflect the relative importance of different aspects in role-playing scenarios. We plan to implement this in the website in the following revisions, potentially allowing for dynamic weighting based on specific use cases or user preferences.
>
> 5. **Single annotator**. We acknowledge the limitations of our current validation approach and are already addressing them. For better reliability, we are expanding our annotation pool to 5 annotators. We have already collected most of these expanded annotations, and we hope to include the updated results in the next paper revision.
>
> 6. **64 conversations per model**. As we stated in the paper, the budget is the reason for having only 64 conversations. We could spend more on the static benchmark, but we are trying to include new models that are constantly appearing.
>
> We appreciate the constructive feedback and plan to incorporate these suggestions in the next revisions, particularly the weighted scoring scheme and the expanded pool of annotators.

---

> > ### Author Response · Authors · 2024-11-29
> >
> > Dear Reviewer u32x, thank you once again for your thoughtful review. With a few days left before closing the author-reviewer discussion period, we wanted to check whether our response and our new revisions have addressed your concerns.
> >
> > From the list above, points 1, 2, 3, and 6 are addressed directly in our comment, point 4 is addressed in the supplementary materials (the website), and point 5 is addressed in the new revision of the paper.
> >
> > If you have any additional questions or feedback, we will gladly discuss them further.

---

### Author Response · Authors · 2024-11-22
**First rebuttal revision**

Dear reviewers,

Thank you for your detailed feedback on our submission. We have made several significant changes to address your comments and improve the paper's quality. Here is a summary of the changes.

### Changes from the original version
1. **Expanded annotation team**: Added 4 new annotators and updated correlation data in Tables 1 and 2, as suggested by reviewers **u32x**, **8Pvv**, **Fuez**, and **ncgD**.

2. **Enhanced annotation documentation**: Added annotation process details to section 4.1 and inter-annotator agreement tables to Appendix B, following **ncgD**'s suggestion.

3. **Text improvements**: Revised Introduction and Related Work sections based on **8Pvv**'s feedback:
- Fixed typos
- Added role-play model applications
- Added introduction citations
- Rephased Related work into 3 subsections instead of 6
- Clarified dynamic setup
- Explained asymmetrical design
- Added key findings to introduction/results
- Relocated version 2 motivation to section 3.3
- Listed annotation models in Appendix B
- Improved example and prompt readability in Appendix A and C

### Supplementary material updates
1. **Weighted scoring**: Added metric weight selector to the website (**u32x**'s suggestion)
2. **Annotation materials**: Added instructions and UI configurations to the repository


### Pending changes
1. Adding Creative Writing benchmark comparison rationale (**ncgD**)
2. Including additional benchmark comparisons (**8Pvv**)
3. Adding analysis of interrogator/user consistency and situation relevance (**oV7o**)

### Things that won't be changed
1. **Metrics**: Original metrics remain.
2. **Sample size**: Maintaining 64 conversations per model due to budget constraints.
3. **Version comparison**: Not possible due to version 1 architecture; explanation is added to the paper.


We believe these changes have significantly strengthened our paper. We look forward to your feedback on the revisions and will address the remaining points in our next update.

---

### Author Response · Authors · 2024-11-26
**Second rebuttal revision**

### Changes from the first rebuttal version
1. Motivation for comparing with the Creative Writing benchmark is added (section 4.3, **ncgD** suggestion).
2. An additional comparison with another role-play benchmark (RPBenchAuto) is added (at the end of section 5, **8Pvv** suggestion).
3. We added a topical analysis of a role-play dataset to show that interrogator situations represent real user intents (Appendix C, **oV7o** suggestion).

### Supplementary material updates
1. We published scripts for creating plots from the paper, topical dataset analysis in the repo, and input and output data for these things.

These changes have improved our paper. We look forward to any additional comments.

---

### Meta-Review · Area_Chair_wjwS · 2024-12-22

**Metareview:**

This paper introduces a benchmark that includes three types of models: a player model that plays the role of a specific character, an interrogator model that mimics the role of an actual user and interacts with the player model, and a judge model that assesses the quality of the conversation between the player and the interrogator. As mentioned by one of the experienced reviewers, such setups are commonly used in other papers these days, such as https://aclanthology.org/2024.acl-long.152/ for task oriented dialogue evaluation, even though there may not be a specific benchmark that is organized in this fashion. Reviewers highlighted strengths of the paper, such as the interesting focus on evaluation of role-playing abilities of LLMs or the codebase that can enable future research. However, they also listed several weaknesses, such as the limited novelty of the work, possible mismatch between actual user and LLM evaluations and small datasets used in the evaluations.

**Additional Comments On Reviewer Discussion:**

Authors provided rebuttals to all reviewers, however, one of the reviewers mentioned their concerns were not fully addressed.

---

### Decision · Program_Chairs · 2025-01-22

Reject